# Understanding the CH_4_ Conversion over Metal Dimers from First Principles

**DOI:** 10.3390/nano12091518

**Published:** 2022-04-29

**Authors:** Haihong Meng, Bing Han, Fengyu Li, Jingxiang Zhao, Zhongfang Chen

**Affiliations:** 1School of Physical Science and Technology, Inner Mongolia University, Hohhot 010021, China; dundun_0521@163.com (H.M.); binghan1214@163.com (B.H.); 2Key Laboratory of Photonic and Electronic Bandgap Materials, College of Chemistry and Chemical Engineering, Ministry of Education, Harbin Normal University, Harbin 150025, China; 3Department of Chemistry, University of Puerto Rico, Rio Piedras Campus, San Juan, PR 00931, USA

**Keywords:** metal dimers, nanozymes, methane conversion, density functional theory

## Abstract

Inspired by the advantages of bi-atom catalysts and recent exciting progresses of nanozymes, by means of density functional theory (DFT) computations, we explored the potential of metal dimers embedded in phthalocyanine monolayers (M_2_-Pc), which mimics the binuclear centers of methane monooxygenase, as catalysts for methane conversion using H_2_O_2_ as an oxidant. In total, 26 transition metal (from group IB to VIIIB) and four main group metal (M = Al, Ga, Sn and Bi) dimers were considered, and two methane conversion routes, namely *O-assisted and *OH-assisted mechanisms were systematically studied. The results show that methane conversion proceeds via an *OH-assisted mechanism on the Ti_2_-Pc, Zr_2_-Pc and Ta_2_-Pc, a combination of *O- and *OH-assisted mechanism on the surface of Sc_2_-Pc, respectively. Our theoretical work may provide impetus to developing new catalysts for methane conversion and help stimulate further studies on metal dimer catalysts for other catalytic reactions.

## 1. Introduction

Global warming is gaining increasing concern worldwide. Greenhouse gases include carbon dioxide, methane, nitrous oxides, and other gases. According to EPA, carbon dioxide accounted for ca. 80% of all greenhouse gas emissions from human activities in the USA in 2019. Though methane has a lower emission (10%), it is also a major greenhouse gas since its greenhouse effect is 21 to 23 times that of carbon dioxide [1]. Therefore, the effective conversion of methane into value-added chemicals (instead of direct burning) is of both environmental and commercial importance [2,3,4].

The direct conversion of methane mainly associates with the high C−H bond strength (~434 kJ/mol) in the non-polar and highly symmetric methane [5,6,7,8]. In traditional industries, methane is first converted into syngas, then transferred to liquid hydrocarbons by Fischer–Tropsch process, which not only causes waste of resources, but also requires a high equipment maintenance cost [9,10,11,12,13,14]. Biological enzyme catalysis may be a good alternative since it has the advantages of high product selectivity and mild reaction conditions: compared with the current industrial process, the direct conversion of methane to methanol that occurs in the methane monooxygenase (MMO) from *Methylococcus capsulatus* is much more efficient [15,16,17,18]. Due to their high catalytic efficiency, biological enzymes can greatly increase the rate of chemical reactions, saving time and cost. Unfortunately, structural instability and sensitivity to the environment greatly limit their performance in industrial applications. A promising way to conquer such challenges is to mimic the MMO with the binuclear active sites [19,20,21]. Zeolites such as ZSM-5 are able to form stable binuclear centers (diiron or dicopper) in similar enzymes, and exhibit unprecedented high performance in methane conversion, but their catalytic mechanism and structure–property relationship remain unclear. The direct conversion of methane at room-temperature and atmospheric pressure is still an unsolved but high-rewarding challenge [22,23].

Nanozymes are a class of nanomaterials with unique enzyme-like properties, which have very similar active sites and catalytic mechanisms to biological enzymes. The first nanozymes were discovered in 2007, since then more than 300 nanomaterials have been found to have enzymatic activity [24,25,26]. Since “single-atom catalysis” was proposed in 2011, the concept of single-atom nanozymes (SAzymes) has also emerged as a research hotspot [27,28,29,30]. Compared with single-atom catalysts, bi-atom catalysts may possess improved catalytic performance [31,32]. For example, Yan et al. showed that Pt dimers embedded in graphene have 17-fold and 45-fold higher catalytic activity for the hydrolytic dehydrogenation of ammonia borane than its corresponding single-atom and nanoparticle counterparts [33]. Li et al. demonstrated that Cu dimers supported on C_2_N layers exhibited superior performance for CO oxidation compared to Cu_1_@C_2_N [34], and showed excellent performance with a small confinement potential of −0.23 V for electrochemical CO_2_ reduction [35]. However, to date, few studies have been reported on the catalytic performance of supported metal dimers for methane conversion. Inspired by the advantages of nanozymes and bi-atom catalysts, we designed a series of supported metal dimer catalysts for methane conversion by mimicking the binuclear centers in biological enzymes based on density functional theory computations.

The two-dimensional (2D) phthalocyanine-based (Pc) catalysts have a high surface area to volume ratio, abundant binding sites for anchoring metal atoms and the ability to prevent these metal atoms from aggregating into clusters. In 2011, Abel et al. [36] successfully prepared FePc complex and characterized the samples using scanning tunneling microscope (STM) at room temperature. Later on, Matsushita et al. [37] synthesized a rectangular phthalocyanine with two adjacent transition metal sites. Since Pc and transition metals are of low-cost, environmentally benign, more readily available than precious metals, the Pc-supported transition metal catalysts can be produced in a low-cost manner. Note that DFT calculations have been widely used to provide guidance in conversion/bonding/adsorption of molecules/clusters and reactions [38,39]. Here, first-principles calculations were conducted to explore the potential of all the 3d, 4d, and 5d non-toxic transition metals and the four main group metal (M = Al, Ga, Sn and Bi) dimers supported on the Pc (M_2_-Pc) for methane conversion.

## 2. Computational Methods

All the computations were carried out by spin-polarized density functional theory (DFT) calculations including van der Waals (vdW) corrections (DFT-D2) [40], as implemented in Vienna Ab initio Simulation Package (VASP) using the projector augmented wave (PAW) method [41]. The generalized gradient approximation (GGA) with the Perdew–Burke–Ernzerhof (PBE) exchange-correlation functional was adopted [42]. The energy cutoff for the plane-wave basis set was chosen as 550 eV, the systemic energy tolerance and the remaining total force were set as 1 × 10^−5^ eV and 0.01 eV Å^−1^, respectively. The Brillouin zone was sampled with a 5 × 5 × 1 k-points grid of the Monkhorst–Pack scheme [43] for geometry optimization, and a denser k-mesh of 15 × 15 × 1 for electronic structure computations. To avoid interactions between periodic images, a vacuum space of 15 Å was used in the perpendicular direction of the 2D layer. The reaction energy barriers were estimated using the climbing-image nudged elastic band (CI-NEB) method [44,45], and the transition states were obtained by relaxing the force below 0.05 eV/Å. The entropic effects were not included in estimating reaction barriers. The binding energy ($E_{b}$) of a metal atom was computed from the following equation.
$$E_{b}=\left(E_{tot}-E_{sub}-2*E_{M}\right)/2$$
where $E_{tot}$,$\;E_{sub}$, and $E_{M}$ represent the total energy of the complex of substrate and metal atoms, the energy of the substrate, and the energy of a free *M* atom, respectively. According to this definition, a more negative *E_b_* value indicates a higher thermodynamic stability. The adsorption energy ($E_{ads}$) of an adsorbate was computed according to the following equation:$$E_{ads}=E_{tot}-E_{cat}-E_{adsorbate}$$
where the $E_{tot}$ is the total energy of an adsorbate adsorbed on the catalyst, $E_{cat}$ and $E_{adsorbate}$ represent the energies of the catalyst and a free adsorbate, respectively. The reaction energy ($E_{rea}$) and activity energy barrier ($E_{act}$) were calculated using the following expressions:$$E_{rec}=E_{FS}-E_{IS}$$
$$E_{act}=E_{TS}-E_{IS}$$
in which $E_{FS}$, $E_{IS}$, and $E_{TS}$ denote the energies of the final, initial, and transition states, respectively. The reaction mechanism can also be effectively modeled by ab initio molecular dynamics (AIMD) simulations at specific temperatures [46].

## 3. Results and Discussion

This work describes our efforts to study the catalytic activity of M_2_-Pc on the conversion of methane. In this paper, we use H_2_O_2_ as the oxidant, because its reaction by-product is only water, which is a green catalyst [47]. Moreover, H_2_O_2_ has been widely used as an oxidant to study the conversion of methane [48,49,50,51,52]. The CH_4_ oxidation with H_2_O_2_ via both *OH- and *O-assisted mechanisms was investigated in detail [48].

### 3.1. Geometric Structure and Stability of M_2_-Pc

First, the geometric structure of the Pc monolayer was optimized, and the lattice parameters *a* and *b* in the Pc monolayer of 14.13 Å were used. As shown in Appendix A, the unique cavity structure of Pc can provide ideal anchoring sites for the metal atoms to be connected, to four isoindole rings, preventing their migrating and aggregating. The computed energy band gap of Pc sheet is 0.94 eV (in Appendix A). The geometric structures and related information of the optimized M_2_-Pc are shown in Appendix A. Obviously, due to the different radii of the metal atoms, the structures of M_2_-Pc are slightly different. The anchored metal atoms with smaller atomic radii form an in-plane configuration in the Pc cavity (Al, Mn, Fe, Co, Ni, Cu, Ga, Ru, Rh, Re, Os, and Ir), while others with larger atomic radii are pulled out of the Pc plane and lead to a buckled structure.

To confirm the stability of metal dimers embedded in the Pc sheet, the binding energy ($E_{b}$) was calculated. Meanwhile, the corresponding metal bulk cohesive energy ($E_{bulk}$) were compared (Appendix A), which are less negative than $E_{b}$, indicating that the interaction between the metal atoms and Pc monolayer is very strong, i.e., the anchoring of metal dimers on the Pc has strong coupling and good stability. We also performed AIMD simulations of the Ag_2_-Pc monolayer, whose binding energy is the least favorable (−4.13 eV) among the considered systems (−13.47~−4.13 eV), and found that the monolayer structure was well kept during 5 ps’s annealing at 300, 800, 1000, 1300, and 1500 K, respectively, and bond breakage occurred at 1500 K (Appendix A). Therefore, all the models in our work have high thermal stability, and in the next section, we will study the catalytic mechanism for methane conversion on these M_2_-Pc catalysts.

### 3.2. Decomposition of H_2_O_2_ on M_2_-Pc

Since the adsorption and dissociation of oxidants are important in methane oxidation, we considered both side-on and end-on configurations of the H_2_O_2_ adsorption (Appendix A) on the examined 30 M_2_-Pc, including 26 transition metals from IB to group VIIIB and four main group metals (M = Al, Ga, Sn and Bi).

Upon adsorption, H_2_O_2_ will spontaneously decompose on the surface of the nine M_2_-Pc (M = Sc, Ti, V, Y, Zr, Nb, Hf, Ta, and W), all of which are highly exothermic (as shown in Appendix A). As shown in Figure 1, H_2_O_2_ can be dissociated when the electronic state of the embedded atoms is a semi-occupied state. Among them, a H_2_O_2_ is spontaneously decompose into a H_2_O and an adsorbed oxygen ($H_{2}O_{2}\to \;*O+*H_{2}O$) on the Nb_2_-Pc with the energy release of −6.08 eV, and into two adsorbed hydroxy groups ($H_{2}O_{2}\to 2*OH$) on the five M_2_-Pc (M = Ti, V, Y, Hf, and Ta with the exothermic energy of −7.51, −6.70, −5.03, −8.71, and −9.04 eV, respectively), while into either a water molecule and an atomic oxygen (by releasing the heat of −3.75, −6.29, and −4.78 eV, respectively), or two OH species (by releasing the heat of −5.19, −8.16, and −7.41 eV, respectively) on the Sc_2_-Pc, Zr_2_-Pc, and W_2_-Pc. All the surface-adsorbed oxo species (*O) occupy the bridge position of the metal dimers. We also examined the magnetic moments and Bader charges on the metal atoms of these nine M_2_-Pc monolayers (M = Sc, Ti, V, Y, Zr, Nb, Hf, Ta, W) (Appendix A), among which the Sc_2_-Pc, Ta_2_-Pc, Y_2_-Pc, and W_2_-Pc have spin states in singlet, the Nb_2_-Pc, Ti_2_-Pc, Zr_2_-Pc, and Hf_2_-Pc are triplet, while V_2_-Pc is in quintet spin state. According to the dissociation structure and energy, H_2_O_2_ dissociation on these nine catalysts (Appendix A) are much more exothermic than the previously reported dissociation reaction of H_2_O_2_ on Pd(111) and Au/Pd(111) surfaces (−1.76 and −1.58 eV for two adsorbed hydroxy groups, and −2.27 and −2.06 eV for absorbed oxygen) [53]. Therefore, it is believed that the decomposition of H_2_O_2_ on these nine catalysts is likely to occur under environmental conditions. Unexpectedly, Fe_2_ and Cu_2_ metal centers, very common active center in biological systems and some biomimetic compounds [54], cannot decompose H_2_O_2_ to form reactive intermediates on Pc sheet.

### 3.3. Catalytic Conversion of Methane on the M_2_-Pc

Previous theoretical studies by Yoo et al. [53] showed that when *OH and *O species exist on the surface, the activation energy barrier of the C−H bond on the precious metal will be decreased [48,55,56]. Based on the spontaneous decomposition of H_2_O_2_ on M_2_-Pc, two mechanisms of CH_4_ conversion will be examined: the *OH-assisted mechanism on Sc_2_-Pc, Ti_2_-Pc, V_2_-Pc, Y_2_-Pc, Zr_2_-Pc, Hf_2_-Pc, Ta_2_-Pc, and W_2_-Pc, the *O-assisted mechanism on Nb_2_-Pc, Sc_2_-Pc, Zr_2_-Pc, and W_2_-Pc.

#### 3.3.1. OH-Assisted Methane Conversion

We first examined the methane conversion over Sc_2_-Pc, Ti_2_-Pc, V_2_-Pc, Y_2_-Pc, Zr_2_-Pc, Hf_2_-Pc, Ta_2_-Pc, and W_2_-Pc via *OH-assisted mechanism. The two quenching reactions of *OH, namely the disproportionation of two OH groups to H_2_O and O $(2\left(*OH\right)\to *O+*H_{2}O$) [57] and the self-reaction of H_2_O_2_ $(H_{2}O_{2}+2\left(*OH\right)\to O_{2}+2H_{2}O$) [58], will result in a low reaction efficiency, thus the quenching reactions were investigated before the calculation.

The disproportionation of two *OH groups to H_2_O and *O on the Ti_2_-Pc, V_2_-Pc, Y_2_-Pc, Hf_2_-Pc, and Ta_2_-Pc were firstly evaluated. As shown in Appendix A, the reaction on Y_2_-Pc, Hf_2_-Pc, and Ta_2_-Pc are difficult (Gibbs free energies are 1.83, 1.72, and 1.63 eV, respectively), in comparison, the free energy change on Ti_2_-Pc and V_2_-Pc are both less than 1 eV. However, the energy barriers on Ti_2_-Pc and V_2_-Pc are as high as 1.95 and 2.59 eV, respectively (Appendix A), indicating that the reaction is also difficult to proceed on these two catalysts. Though on Sc_2_-Pc, Zr_2_-Pc, and W_2_-Pc catalysts, the decomposition of H_2_O_2_ into a H_2_O and an adsorbed oxygen are more energy-efficient than the splitting into two *OH, considering that the splitting into two *OH on these three catalysts is spontaneous, we also calculated the methane conversion via *OH-assisted mechanism on these three catalyst surfaces.

The formation of O_2_ is also a competitive reaction, which means that H_2_O_2_ may become its own scavengers. The energy parameters for the self-reaction on the six M_2_-Pc (M = Ti, V, Zr, Sc, Y, and Hf) are given in Appendix A, the rather high Gibbs free energies for the reaction on the Ti_2_-Pc, V_2_-Pc, and Zr_2_-Pc (2.61, 2.69, and 3.08 eV, respectively) indicate that the self-reaction on these three catalysts is not thermodynamically favorable. No matter what initial structure is built on the Ta_2_-Pc and W_2_-Pc, they will become two adsorbed hydroxy groups after relaxation, i.e., the reaction will not proceed on either Ta_2_-Pc or W_2_-Pc. The calculated reaction energies over the Sc_2_-Pc, Y_2_-Pc, and Hf_2_-Pc are 0.54, 0.65, and −0.18 eV, respectively; however, the activation energy barriers are 2.27, 1.66, and 1.36 eV, respectively (Appendix A), which means that the *OH is also difficult to quench on these three catalysts.

The above results showed that *OH will not be quenched on the eight catalysts examined in this section. Based on this finding, we investigated the methane conversion reaction via *OH-assisted mechanism.

First, CH_4_ is weakly adsorbed on the catalyst covered by OH, and the C−M bond lengths are 2.59, 2.42, 2.37, 2.81, 2.51, 2.45, 2.64, and 2.42 Å on the Sc_2_-Pc, Ti_2_-Pc, V_2_-Pc, Y_2_-Pc, Zr_2_-Pc, Ta_2_-Pc, W_2_-Pc, and Hf_2_-Pc, respectively. The adsorption energies are −0.15, −0.53, −0.52, −0.41, −0.49, −0.61, 0.28, and −0.22 eV, respectively. The reaction energies of surface *OH groups attracting H from CH_4_ are 0.76, 0.81, 0.59, 0.95, −0.14, 0.58, and 0.83 eV on Sc_2_-Pc, Ti_2_-Pc, V_2_-Pc, Y_2_-Pc, Zr_2_-Pc, Ta_2_-Pc, and Hf_2_-Pc, respectively; the corresponding activation energy barriers are 1.09, 0.85, 1.41, 1.35, 0.86, 0.62, and 1.22 eV. The reaction of extracting H from CH_4_ on the W_2_-Pc is not considered because its repulsiveness to CH_4_ (Appendix A). These analyses suggested that Ti_2_-Pc, Zr_2_-Pc, and Ta_2_-Pc can catalyze the C-H breakage due to favorable reaction energies and relatively mall activation barriers (0.85, 0.86 and 0.62 eV). Note that though the activation barrier on Sc_2_-Pc is slightly high (1.09 eV), considering that the O-assisted methane conversion on this catalyst benefits from the assistance of *OH (see Section 3.3.2), we also investigated the the *OH-assisted mechanism on Sc_2_-Pc. Thus, four catalysts which are feasible to break the C-H bonds, namely, Sc_2_-Pc, Ti_2_-Pc, Zr_2_-Pc, and Ta_2_-Pc, will be further investigated.

Then, we calculated the subsequent reactions over Sc_2_-Pc, Ti_2_-Pc, Zr_2_-Pc, and Ta_2_-Pc. Figure 2a–d summarizes the corresponding potential energy profile and the optimized geometries along the reaction path on these four M_2_-Pc. After the cleavage of the C−H bond, CH_3_ and H_2_O are adsorbed on the four catalyst surfaces. Next, the desorption of *CH_3_, the desorption of *H_2_O, and the reaction with OH in the solution to generate CH_3_OH were considered, respectively. The best path for all these four catalysts is to react with OH in the solution. Note that *CH_3_ will react with ∙OH in the solution to form CH_3_OH on Sc_2_-Pc and Ti_2_-Pc, the reaction proceeds spontaneously on Sc_2_-Pc, and the energy barrier on Ti_2_-Pc is only 0.08 eV. While on Zr_2_-Pc and Ta_2_-Pc, OH will combine with the H atom of *OH to generate H_2_O instead of reacting with *CH_3_ (this process occurs spontaneously on both catalysts). After desorbing H_2_O, *CH_3_, and *OH will remain on the surface. Unexpectedly, on both Zr_2_-Pc and Ta_2_-Pc catalysts, *CH_3_ will not combine with *OH on the surface to form CH_3_OH, but will combine with ∙OH in the solution to generate CH_3_OH (the energy barriers are 0.84 and 0.08 eV, respectively, as shown in Figure 2c,d).

To summarize, we identified that four catalysts, namely, Sc_2_-Pc, Ti_2_-Pc, Zr_2_-Pc, and Ta_2_-Pc, show high CH_4_ conversion activity with the assistance of *OH. The rate-limiting steps of Sc_2_-Pc, Ti_2_-Pc, and Zr_2_-Pc are the cleavage of the first C−H bond, with energy barriers of 1.09, 0.85, and 0.86 eV, respectively. The rate-limiting steps of Ta_2_-Pc is the extraction of ∙OH in the solution, with energy barriers of 1.11 eV.

#### 3.3.2. O-Assisted Methane Conversion

Then we examined *O-assisted methane conversion on the Nb_2_-Pc, W_2_-Pc, Zr_2_- Pc, and Sc_2_-Pc catalysts. The adsorption energies of *H_2_O on these four M_2_-Pc are 0.27, 0.13, 0.21, and 0.31 eV, respectively, indicating the feasibility to desorb *H_2_O from these slabs to form O-adsorbed catalysts. According to previous studies, the C−H bond cleavage may occur via either a surface-stabilized ($*O+CH_{4}\to \;*OH+*CH_{3}$) or a radical-like mechanism ($*O+CH_{4}\;\to \;*OH+\cdot CH_{3}$) [59,60]. Appendix A shows the energy diagram of methane conversion on these four *O-adsorbed catalysts following the surface-stabilized or radical-like mechanism, and the competition reaction pathway of the *CH_3_ dehydrogenation is also considered ($*CH_{3}+*OH\to *CH_{2}+*H_{2}O$). In contrast to the quintet state of Fe(IV)(oxo) in Fe(IV)O/MOF-74 [61], the M-O-M moiety (M = Sc, Zr, Nb, W) is singlet in the ground state, since the magnetic moment on either M or O is zero. Accordingly, the oxidation state of O and Sc/Zr/Nb/W could be assigned as −2 and +3/+4/+4/+4, respectively, which is quantitatively in line with our Bader charge analysis (Appendix A). Notably, the spin state of Nb_2_-Pc and Zr_2_-Pc switches from the triplet to the singlet when forming M-O-M moiety, agreeing well with our recent theoretical observations in metal dimer-related catalysis [62]. The partial density of states (PDOS) of CH_4_ adsorption on M-O-M moiety (Appendix A) shows that there is orbital hybridization between M-O-M and CH_4_.

Among these examined catalysts, *O-assisted conversion of methane is energetically more favorable on the Sc_2_-Pc monolayer through a surface-stabilized mechanism (Appendix A). This is different from the traditional single-site catalyst, which prefers the free radical mechanism [60,61,62,63,64,65]. The potential energy profile and the reaction path of *O-assisted methane C−H bond cleavage on the Sc_2_-Pc surface are illustrated in Figure 3, in which the C−H bond activation is found as the rate-determining step for the first methanol formation. The reaction begins with the adsorption of CH_4_, which is physically adsorbed on the O-preadsorbed Sc_2_-Pc through van der Waals interaction as the initial state with the adsorption energy of −0.09 eV, where the C−H bond length is 1.10 Å, slightly longer than that in the free CH_4_ molecule (1.07 Å), and the distance between O and H is 1.84 Å. In the transition state, the distance between C and H is elongated to 1.47 Å, and the distance between O and H is shortened to 1.21 Å, both of which are between the initial state and the final state. In the final state, *OH and *CH_3_ will form one Sc−C (bond distance of 2.25 Å) and two Sc−O bonds (bond lengths 1.97 and 2.37 Å) on surface. In other words, the *O-assisted methane activation on the Sc_2_-Pc catalyst follows the surface-stabilized mechanism, and the energy barrier for activation of the first C−H bond is 0.63 eV, which is much lower than the *O-assisted Au (111) surface (1.33 eV) [53].

After the C-H bond cleavage on Sc_2_-Pc, the position of *O is transferred from the bridge site of two Sc atoms to the top of one Sc atom, and *CH_3_ is adsorbed on the other Sc atom. Unlike single-site active center catalysts, bi-atom active centers increase the adsorption strength of the intermediates, thereby preventing the combination of *CH_3_ and *OH, which can be seen from the very high energy barrier (1.65 eV) in Figure 3.

We also considered that *CH_3_ reacts with ∙OH in the solution to form *CH_3_OH. This reaction proceeds spontaneously (the initial state structure and the final state structure are shown in Appendix A) by releasing energy of 0.54 eV (the blue line in Figure 3). The desorption of *CH_3_OH requires 0.13 eV of energy. After desorbing CH_3_OH, *OH remains being adsorbed on the surface occupying the bridge position of Sc dimers. Subsequently, we investigated two reaction paths: *OH moves to the top site of Ta atom, and *OH on the bridge position continues to activate CH_4_. As shown in Figure 3, the C−H bond cleavage assisted by the *OH on the bridge site is endothermic by 0.87 eV, while the migration of the *OH on the bridge site to the top site is slightly endothermic by 0.23 eV. Therefore, the *OH prefers moving to the top site, and the subsequent reaction is the same as discussed in Section 3.3.1. Noted that the active site motif for H_2_O_2_-converting methane is M-O-M (where M is a metal center), different from the biological MMOs containing binuclear Fe centers for the oxidation of CH_4_ to CH_3_OH by O_2_, where the reaction involves the formation of a pair of highly active iron(IV)oxo groups in a “diamond core” arrangement [61].

Note that the DFT self-interaction errors can have significant effects on the reactivity of high-valent Fe species during the oxidation of methane in metal-organic frameworks [66]. Thus, we tested the reaction of the first C–H bond cleavage using the HSE06 functional [67] and compared with the PBE result, Ti_2_-Pc was taken as a representative due to its best catalytic performance among the catalysts examined in this work. We found that reaction barriers (0.84 vs 0.85 eV) and reaction energies (0.78 vs 0.81 eV) are very close from the two methods (Appendix A). Thus, we conjecture that PBE results are reliable for our systems, and we adopted PBE functional through our calculations.

## 4. Conclusions

To sum up, we designed low-cost bi-atom (M_2_-Pc) catalysts for CH_4_ conversion using a two-dimensional material Pc to support the metal dimers. Two methane conversion routes, namely *O-assisted and *OH-assisted mechanisms, over M_2_-Pc were systematically studied by means of density functional theory computations. Our computations identified four high-performance catalysts for methane conversion: the Sc_2_-Pc surface following a combined *O-assisted and *OH-assisted mechanism with the C−H bond breaking energy barrier of 0.63 eV, and Ti_2_-Pc, Zr_2_-Pc, and Ta_2_-Pc following *OH-assisted mechanism with energy barriers of 0.85, 0.86, and 1.11 eV, respectively, all these activation barriers are lower than that on the Au(111) surface (1.33 eV) [53]. This work clearly demonstrated that the M_2_-Pc monolayers can serve as low-cost and efficient bi-atom catalysts for methane conversion, which not only enrich the family of bi-atom catalysts, but also provides new strategy to design effective bi-atom catalysts for methane conversion and related reactions.

## Figures and Tables

**Figure 1 nanomaterials-12-01518-f001:**
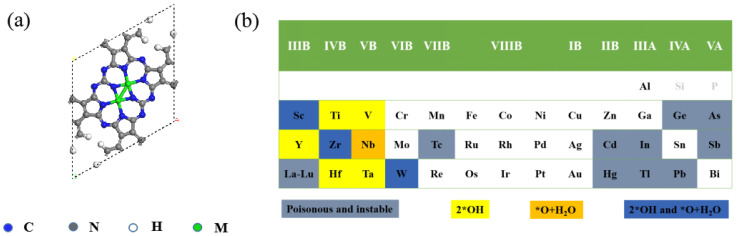
(**a**) Top view of M_2_-Pc sheet. (**b**) The metals examined in this work. The poisonous metals were indicated in gray; the metals assist H_2_O_2_ decomposition into 2*OH or *O and H_2_O are highlighted in yellow and orange, respectively; the metals could decompose H_2_O_2_ into either 2*OH or *O and H_2_O are represented in blue; metals in white cannot spontaneously decompose H_2_O_2_.

**Figure 2 nanomaterials-12-01518-f002:**
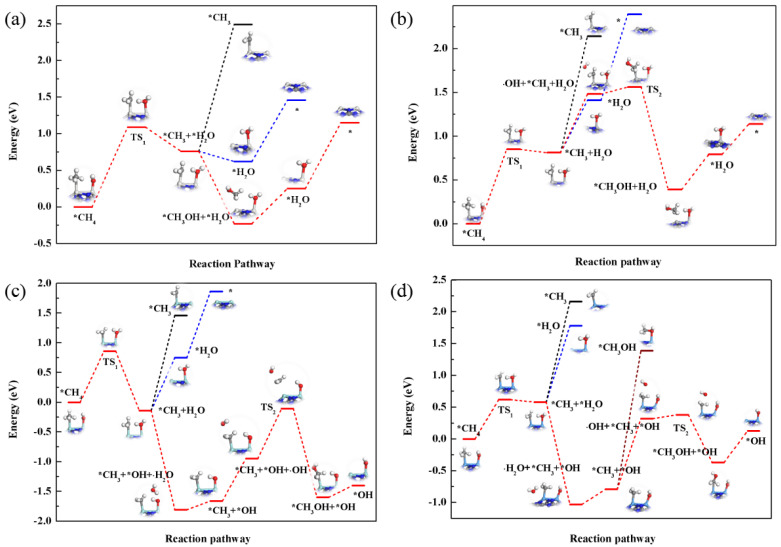
Reaction pathway of *OH-assisted CH_4_ decomposition on the Sc_2_-Pc (**a**), Ti_2_-Pc (**b**), Zr_2_-Pc (**c**), and Ta_2_-Pc (**d**) monolayers. Blue, black, and red lines represent the reaction paths for *CH_3_ desorption, *H_2_O desorption, and the reaction with OH in the solution, respectively.

**Figure 3 nanomaterials-12-01518-f003:**
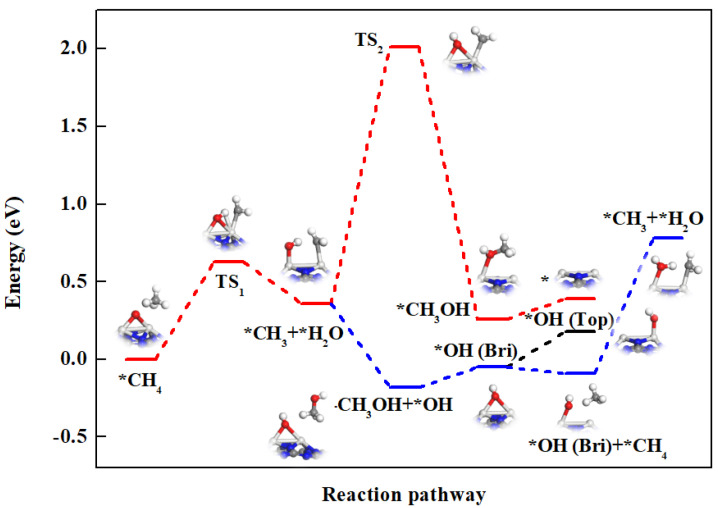
Reaction pathway of *O-assisted CH_4_ decomposition on the Sc_2_-Pc monolayer. Blue, black, and red lines represent the three paths of *CH_3_ reacting with *OH, *CH_3_ reacting with ∙OH in solution, and the migration of *OH from bridge to top site, respectively.

## Data Availability

The data presented in this study are available upon reasonable request from the corresponding authors.

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
