# Peer review of "Understanding the CH4 Conversion over Metal Dimers from First Principles"

_nanomaterials, 2022, doi:10.3390/nano12091518_

Round 1

Reviewer 1 Report

This manuscript is dedicated to studying the potential of metal dimers embedded in phthalocyanine monolayers (M2-Pc) as catalysts for methane conversion, employing the DFT methodology. The authors studied a total of 26 transition metal (and 4 main group metal (M = Al, Ga, Sn and Bi) dimers while two methane conversion routes, namely *O-assisted and *OH-assisted, were tested. The work offers a sophisticated and well-planned setup of model systems, adequate level of theory, and a deep analysis to achieve better understanding of how exactly the methane conversion proceeds. From practical point of view, the reported results bring new knowledge about developing new catalysis for methane conversion which certainly is an original contribution in the present context.

Thus, the ambitious task in this work covers an array of hot topics of research of the complex aspects related to metal dimers catalysts for catalytic reactions with wide perspectives for groundbreaking applications that are currently attracting much research interest.

The authors chose an adequate structure of the manuscript – an excellent point of departure for such a study. Finally, the authors provided a balanced realistic and nicely illustrated presentation of their numerical results and corresponding analysis that is of much scientific and practical interest and adds to new knowledge to the field.

In my opinion, the fine detailing in the present work, the insightful and balanced discussion of the results, as well as the very good figures, permit competent readers to utilize the manuscript as a guidance for future work. Consequently, this manuscript presents an efficient and beneficial basis for promoting and solving next step challenges in this field.

Moreover, the manuscript benefits from a clear motivation and it is an easy and informative read. The manuscript is also excellent in terms of clarity and accuracy of language.

The present manuscript is a significant contribution, this work once published would be quite useful as well as instructive and suggestive in terms of further studies and to a wider readership.

There are some minor issues with this already excellent manuscript that will need to be addressed before becoming suitable for publication, i.e., it can be considered for publication after a minor revision:

1: The authors miss part of bigger picture of different routes of conversion /bonding/adsorption of molecules/clusters and reactions whereby also DFT calculations have been widely used for accurate guidance, e.g., Journal of Physics D: Applied Physics 48 (2015) Article number 295104, Thin Solid Films 515 (2006) Pages 1192 – 1196. Such works approaches and treating are supportive to the credibility of the findings reported in the present manuscript.

2: Are the authors aware of any studies addressing thermal stability of the metal dimers they consider in the present study, at which temperature they break, etc.? Shortly discussing this aspect may be of practical interest for the systems studied.

3: It would be informative and interesting to submit to ab initio molecular dynamics calculations (AIMD) tests some of the reaction pathways addressed here, also because of explicitly including temperature. Are the authors aware of any such efforts for similar systems? I’m not suggesting that the authors need to carry out AIMD calculations for the present work.

4: Spell-check and stylistic revision of the paper are still necessary. Some long sentences, misspellings, etc., still are noticeable throughout the text.

Author Response

Dear Referee,

Thank you for your comments to our manuscript submitted to Nanomaterials (Manuscript ID: nanomaterials-1676240). In the attached file please find our response. For your convenience, all changes in the manuscript are highlighted in red, and the responses are highlighted in blue.

Reviewer 2 Report

See the attachement

Author Response

Thank you for your comments to our manuscript submitted to Nanomaterials (Manuscript ID: nanomaterials-1676240). In the attached file please find our response. For your convenience, all changes in the manuscript are highlighted in red, and the responses are highlighted in blue.

Reviewer 3 Report

The manuscript of Meng et al. describes an intriguing computational study of the methane oxidation potential of a rich selection of metal-atom dimers supported by a phthalocyanine monolayer in the presence of H2O2. This work aims to develop a mechanistic model of methane activation processes in these systems, which combine features of nano-enzymes with those of catalytically active bi-atom catalysts. Density functional theory (DFT) calculations on periodic samples are used to study the energetics and reaction profiles of the decomposition of H2Oof the phthalocyanine-supported metal dimers and of the subsequent O- or OH-driven activation of methane. The results indicate that some of the species examined can initiate the conversion of methane into methanol with energy barriers sufficiently low for them to occur in experimental conditions. 

On the whole, this is a very interesting study into the catalytic activity of metal dimers in hydrocarbon oxidation. The work is focussed, well carried out and the conclusions are convincing. I think this paper provides a very valuable exploratory assessment of the catalytic potential of the systems studied in the activation of methane, one of the most ambitious and sought after current goals of organic and industrial chemistry. In my opinion, this paper deserves publication, but I would like the authors to address the following points before recommending publication.

1) In the Introduction, the authors mention the case of methane monooxygenase (MMO) as a well known example of methane-to-methanol conversion enzymes that exploit di-nuclear metal centers (Fe, in this specific case) to carry out their catalytic reactions. There is an important difference, however, between MMO and the systems studied in this work, namely that MMO uses di-oxygen (rather than H2O2) to oxidize methane, and the reaction involves the formation of a pair of highly reactive iron(IV)oxo groups in a "diamond core" arrangement. I think this point should be noted in the paper, to avoid confusion with the very extensive literature on iron(IV)oxo species in biological and inorganic systems.

2) The authors examine the oxidation of methane via two potential mechanisms, an O-based and an OH-based reaction path. From Figure 3, it looks like the first mechanism relies on the formation of an M-O-M moiety (where M is a metal center). What is the oxidation state of the O atom and of the M atoms, and how does this situation compare to the more "classical" iron(IV)oxo electronic structure? It would very helpful to have some indications on the spin states and orbital energies in this case, and to draw some comparisons with other methane-activating systems.

3) The calculations were carried out using the PBE exchange-correlation functional. It has recently been pointed out (see, e.g., Phys. Chem. Chem. Phys., 2020,22, 12821-12830) that DFT self-interaction errors can have dramatic effects on the reactivity of high-valent Fe species during the oxidation of methane in metal-organic frameworks. It is possible that similar problems would be observed in the metal-phthalocyanine systems studied in this work, because of the coexistence of extended phthalocyanine electronic states and localized metal states. The use of PBE could then result in large errors in the calculated reaction barriers. This should be mentioned in the paper and, if possible, calculations for at least some of the systems at a higher level of DFT (e.g., HSE06) should be carried out to confirm the results. 

4) From the results summarized in Figure 1, it looks like Fe, which is a very common catalytic center in biological systems and in some biomimetic compounds, is not able to decompose H2Oto form a reactive intermediate. I think this is an interesting results, which should be given more emphasis in the text. Obviously, Fe would be a far preferable metal than many others, in view of its abundance and low environmental impact. 

5) Lines 136-138. The sentence "It can be seen from Figure 1 that H2O2 can be dissociated when the electronic state of the embedded atoms is a semi-occupied state." should be clarified. It would also be helpful to have an idea of the total spin of the systems examined. How was the ground spin state determined in each case?

6) Line 35-36. Methanol is labelled a "liquid hydrocarbon". It should be referred to as an "alcohol" instead.

7) Line 41. Replace "methylococcus capsulatum" with "methylococcus capsulatus". 

8) Were entropic effects included in the estimates of the reaction barriers? This should be mentioned.

Author Response

(The authors gave the same response as above.)

Round 2

Reviewer 3 Report

The authors have addressed carefully and satisfactorily the suggestions for improvements from the report. I recommend publication of the revised version of the paper.